# Can a Sacrococcygeal Epidural of 0.25% Bupivacaine Prevent the Activation of the Sympathetic Nervous System during Feline Ovariectomy?

**DOI:** 10.3390/ani14121732

**Published:** 2024-06-08

**Authors:** João Martins, António Eliseu, Sónia Campos, Lénio Ribeiro, Pablo Otero, Patrícia Cabral, Bruno Colaço, José Diogo dos-Santos

**Affiliations:** 1Faculty of Veterinary Medicine of Lisbon, Lusófona University-Lisbon University Center, 1749-024 Lisboa, Portugal; apintoeliseu@gmail.com (A.E.); p5201@ulusofona.pt (S.C.); p4833@ulusofona.pt (L.R.); p6401@ulusofona.pt (P.C.); jose.diogo.agb@gmail.com (J.D.d.-S.); 2Veterinary and Animal Research Center (CECAV), Lusófona University-Lisbon University Center, 1749-024 Lisboa, Portugal; bcolaco@utad.pt; 3Associate Laboratory for Animal and Veterinary Sciences (AL4AnimalS), 1300-477 Lisbon, Portugal; 4Centre for the Research and Technology of Agro-Environmental and Biological Sciences (CITAB), Universidade de Trás-os-Montes e Alto Douro (UTAD), 5000-801 Vila Real, Portugal; 5Department of Anesthesiology and Pain Management, Facultad de Ciencias Veterinarias, Universidad de Buenos Aires, Buenos Aires C1427CWN, Argentina; potero@fvet.uba.ar

**Keywords:** cat, ovariectomy, sympathetic nervous system, sacrococcygeal epidural, parasympathetic tone activity

## Abstract

**Simple Summary:**

This study aimed to assess the efficacy of a sacrococcygeal epidural (ScE) of 0.25% bupivacaine in preventing sympathetic nervous system activation on feline ovariectomy. During the intraoperative period, traditional hemodynamic variables (heart rate, systolic and median blood pressure, and respiratory rate) and Parasympathetic Tone Activity monitoring were used to assess the activation of the sympathetic nervous system after ScE of 0.25% bupivacaine. This study indicates that implementation of ScE reduces the rise of common hemodynamic variables but does not prevent sympathetic nervous system activation during feline ovariectomy.

**Abstract:**

The ovariectomy (OVE) procedure can trigger somatosensory and visceral peritoneal nociception. Sacrococcygeal epidural (ScE) anesthesia may complement or replace systemic analgesia used for feline OVE, reducing opioid consumption and their related undesirable adverse effects and consequently reducing or completely blocking the sympathetic nervous system activation during this procedure. The present study aimed to evaluate the activation of the sympathetic nervous system resulting from adding an ScE injection of bupivacaine 0.25% (0.3 mL kg^−1^) in feline OVE and identify whether this translates to hemodynamic variables stability. A Parasympathetic Tone Activity (PTA) monitor was applied given that it performs analysis of heart rate variability (HRV) detecting changes in sympathetic and parasympathetic tone, making it a good tool for detecting activation of the sympathetic nervous system during the study. Two groups of animals were evaluated in five perioperative times, namely, the control group (CG) (n = 18) with systemic analgesia alone and the sacrococcygeal epidural group (ScEG) (n = 20) with 0.25% bupivacaine combined with systemic analgesia. Thirty-eight female cats were selected. All animals assigned to CG and ScEG were premedicated with dexmedetomidine (20 μg kg^−1^ IM) and methadone (0.2 mg kg^−1^ IM). General anesthesia was induced with propofol IV ad effectum and maintained with isoflurane in 100% oxygen. Heart rate, non-invasive systolic and median blood pressure, respiratory rate, and instantaneous parasympathetic tone activity were recorded. Compared to systemic analgesia alone (CG), sacrococcygeal epidural (ScEG) reduced the rise of common hemodynamic variables but did not prevent sympathetic nervous system activation.

## 1. Introduction

During ovariectomy (OVE), celiotomy, traction, and ligation of the ovarian pedicles and uterine horns can trigger somatosensory and visceral peritoneal nociception [1]. To maintain hemodynamic stability during general anesthesia and undesirable autonomic responses, adequate analgesia should be provided during the intraoperative period [2,3].

The common analgesic protocol used for OVE is based on the combination of systemic opioids, sedatives, and non-steroidal anti-inflammatory drugs [4], which may not be enough to ensure the prevention of intraoperative nociception [5] and sympathetic nervous system activation [2,6].

Sacrococcygeal epidural (ScE) anesthesia is a locoregional technique that prevents perioperative nociception [1,3] and can complement or replace systemic analgesia in cats [1,2]. This technique in cats is safer than lumbosacral epidural since it reduces the risk of piercing of the meninges and adverse effects related to the volume of the administered solution in inadvertent intrathecal injections, given that in cats the spinal cord ends more caudally and the dural sac terminates at the level of the third sacral vertebra [1,2,7,8,9,10]. The epidural application with 0.3 mL kg^−1^ of volume at this level has already been shown to have analgesic advantages in spaying female cats [1,3].

An adequate preventive perioperative nociception can be evaluated by monitoring changes in blood pressure (BP) and heart rate (HR) [11]. However, BP and HR are not objective indicators of nociception and may be subject to confounding factors and not always accurately reflect nociception [12].

In human medicine, nociception monitors are being increasingly used during surgery, and to a lesser extent in veterinary medicine [13]. Nociception monitors such as the Analgesia Nociception Index (ANI) monitor in human medicine [11] and the Parasympathetic Tone Activity (PTA) monitor [13,14] in veterinary medicine analyze the interval between the R-R waves of the ECG. Changes in R-R intervals, also known as heart rate variability (HRV), are influenced by changes in sympathetic and parasympathetic tone [11,13], making them a tool for detecting activation of the sympathetic nervous system. The activation of the sympathetic nervous system has been linked to nociception and, thus, the number provided by the monitor is proposed as a surrogate of a nociceptive response.

In anesthetized patients, PTA values that fall within the range of 50 to 70 indicate the absence of nociception [13,15,16]. PTA values nearing 100 denote a predominant parasympathetic tone or opioid overdose, while values below 50 denote a predominant sympathetic tone associated with elevated stress or nociception [15,17]. PTA monitoring could, therefore, constitute a good tool to investigate the effectiveness of anesthetic techniques such as ScE to control sympathetic nervous system activation in cats [18].

To the authors’ knowledge, this is the first study to report PTA monitoring in ScE with the administration of 0.25% bupivacaine for analgesia during OVE.

This study aimed to evaluate the PTA in cats submitted to OVE receiving an ScE as part of the intraoperative analgesic protocol. We hypothesized that administration of ScE would maintain the parasympathetic tone activity between 50 and 70 during the surgical procedure.

## 2. Materials and Methods

### 2.1. Animals

This study was conducted in accordance with national legislation at the Veterinary Teaching Hospital, Faculty of Veterinary Medicine, Lusófona University in Lisbon and was approved by the Comissão de Ética e Bem-Estar Animal (CEBEA) from the Faculty of Veterinary Medicine of the Lusófona University (15-2022). Informed owner consent was obtained.

A priori power calculation was conducted using the G*Power 3.1.9.7 statistical package program. To achieve 80% power for detecting a medium effect at a significance criterion of α = 0.05 error level, a minimum of 18 cats per group was suggested. This study was blinded, controlled, and randomized using a random sequence generator “https://www.random.org (accessed on 20 December 2022)”. With randomization, 18 animals were allocated in a control group (CG) and 20 animals in a sacrococcygeal epidural group (ScEG).

Female cats assigned with an American Society of Anesthesiologists (ASA) classification status of I underwent elective ovariectomy. All animals were feral or stray domestic shorthair adult queens captured the day before the procedure and food but not water was withheld the night before. Cats were transported to the hospital on the following day in individual cages and were kept in a quiet room.

To determine the weight of the cats before drug administration, the cages were weighed before the cats’ capture and transportation. Pregnant cats or those with changes in physical examination, serum biochemistry, or hematology were excluded.

### 2.2. Blind Study

For this study, three researchers were involved in anesthesia and data collection. The first researcher, an experienced anesthetist (JDS), was responsible for drug preparation, premedication, induction, orotracheal intubation, and ScE but not involved in data collection. The second researcher, a staff member, was involved in transportation to the operating room, positioning of cats, connection to the anesthetic machine, and surgical field antisepsis (AE). The third researcher, an anesthetist, was responsible for monitoring, rescuing decisions, and data collection, without any communication with the first and second investigators (LR). Two experienced surgeons (JM and PC), also blinded, performed the OVE surgeries.

### 2.3. Surgical Technique 

The cats were positioned in dorsal recumbency and a median laparotomy caudal to the umbilicus was performed [19]. The uterus horn was found by digital palpation [19]. After ligation of the ovarian pedicle, two clamps were placed on the proximal uterine horn close to the ovary and a ligature was placed distally [19]. The ovary was removed after cutting between the hemostats and the procedure was repeated in the opposite ovary. After examining the abdomen for bleeding, the ventral linea alba and the subcutaneous tissue were closed using an absorbable suture material in a simple continuous pattern. The skin was closed using an absorbable suture material with an intradermal suture [19].

### 2.4. Anaesthetic Management

Cats assigned to CG and ScEG were premedicated via intramuscular (IM) injection with dexmedetomidine (20 μg kg^−1^; Dexdomitor 0.5 mg mL^−1^, Orion, Finland) and methadone (0.2 mg kg^−1^; Semfortan 10 mg mL^−1^, Dechra, Torino, Italy). All cats rested in their cages until a sufficient level of sedation was achieved. Physical examination and blood samples collected from the jugular vein for serum biochemistry and hematology were performed. A catheter was placed in a cephalic vein.

After premedication, intravenous (IV) fluid therapy (3–5 mL kg^−1^ h^−1^) with lactated Ringer’s solution (Lactated RingerVet; B. Braun Medical Inc., Melsungen, Germany) was provided until the end of surgery. General anesthesia was induced 15 min after premedication and preoxygenation with IV administration of propofol (Propofol 10 mg mL^−1^; B. Braun Medical Inc., Melsungen, Germany) to achieve the desired effect. After the loss of consciousness, the cats’ trachea was intubated with a 3.5 to 4.5 mm cuffed endotracheal tube after desensitization of the larynx with topical application of 0.1 mL of 2% lidocaine (Lidocaine 20 mg mL^−1^, B. Braun Medical Inc., Melsungen, Germany). General anesthesia was maintained with isoflurane (IsoFlo; Zoetis, Madrid, Spain) in oxygen delivered through a non-rebreathing system (FIO_2_ 100%, fresh gas flow of 250 mL kg^−1^ min^−1^) in spontaneous breathing. The end-tidal isoflurane target concentration was set to 1 ± 0.1% for the intraoperative period to maintain the surgical plane of anesthesia. After the onset of general anesthesia, the eyes were lubricated. To minimize temperature loss, a heating blanket (Thermal Blanket Carbonvet cage, B. Braun Medical Inc., Shanghai, China) was used.

### 2.5. Sacrococcygeal Epidural Anesthesia Technique

All cats from the ScEG were placed in a sternal position. After clipping, the sacrococcygeal region was aseptically prepared. The pelvic limbs were pulled forward and the tail hanging. The intervertebral space between the last sacral and the first coccygeal vertebrae was identified by palpation. A 22-gauge, 55 mm insulated needle (Lococare; Belphar, Sintra, Portugal) with an extension line prefilled with 0.25% bupivacaine was used, connected to a peripheral nerve stimulator (NS). The NS’s positive lead was placed on the right thigh’s caudal skin. The NS was set to a fixed electrical current of 0.7 mA, frequency of 2 Hz, and pulse of 0.1 ms [9]. The needle was inserted at a 30° angle toward the tail to reach the intervertebral junction between the last sacral and the first coccygeal vertebrae. The needle was advanced gently until the lateral movement of the tail was detected. The epidural injection was confirmed through the color flow Doppler test (CFDT) performed at the lumbosacral space [9], administering 0.2 mL of saline. Cats that showed a tail motor response and a positive CFDT were then administered 0.3 mL kg^−1^ of 0.25% bupivacaine over 45 s.

### 2.6. Intraoperative Nociception Assessment and Rescue Analgesia

During anesthesia, heart rate (HR; beats minute^−1^), respiratory rate (*f*_R_; breaths minute^−1^), end-tidal carbon dioxide (PE CO_2_; mmHg), inspiratory fraction of carbon dioxide (mmHg), esophageal temperature (TOES), oxygen hemoglobin saturation (SpO_2_; %), and end-tidal isoflurane (EtIso, %) were monitored. Non-invasive blood pressure was performed with an oscillometric method, and a 2.5 cm cuff was placed around the right or left antebrachium. The systolic arterial pressure (SAP) and mean arterial pressure (MAP) were measured at 3-minute intervals. Parameters were monitored by an anesthesia monitor (BeneVision N15; Mindray, Shenzhen, China). During the surgical procedures, hands-on anesthesia was performed by evaluating the eyeball position, jaw tone, and palpebral reflex; all animals were maintained in a surgical anesthetic plane.

To measure the instantaneous PTA (PTAi), the ECG signal was used after placing three flattened crocodile clips moistened with gel on the skin at the level of the olecranon of the thoracic limbs and over the patellar ligament of the pelvic limb [20].

In case of hypotension during anesthesia, defined as SAP < 90 mmHg or MAP < 60 mmHg, a step-based approach was initiated with a 20% reduction of isoflurane, 10 mL kg^−1^ lactated Ringer´s solution within 10 min. Ephedrine (0.1 mg kg^−1^; Labesfal, Tondela, Portugal) or atropine (0.2 mg kg^−1^; B. Braun Medical Inc., Melsungen, Germany) was administered IV when required. During surgery, rescue analgesia was implemented if two of three variables (HR, MAP, or *f*_R_) were increased > 20% from the values recorded before incision. Rescue analgesia consisted of an IV bolus of fentanyl (2 μg kg^−1^; Fentadon; 50 µg mL^−1^, Dechra, Italy).

At the end of general anesthesia, HR, *f*_R_, oral mucous membrane color, and temperature were monitored before recovery in a clean and individual cage. After extubation, meloxicam (0.2 mg kg^−1^; Metacam, 2 mg mL^−1^, Boehringer Ingelheim Vetmedica Gmbh, Ingelheim, Germany) was administered subcutaneously for postoperative analgesia. All cats were discharged on the surgery day and were released the day after surgery. 

### 2.7. Data Collection 

Five surgical times were defined for both groups. Data from outcome variables HR, *f*_R_, SAP, MAP, and PTAi were obtained before skin incision (baseline), at skin incision (T1), at first ovary (T2) and second ovary (T3) removal, and finally at abdominal closure (T4). After the application of surgical stimuli (T1, T2, T3, and T4), the data collected were the PTAi baseline and the maximum value obtained for the hemodynamic parameters (HR, SAP, MAP, and *f*_R_). Before the surgeon proceeded to the next surgical stimulus, it was expected that the PTAi would return to values between 50 and 70. Surgery commenced only after 15 min following epidural injection. Rescue analgesia was recorded as “yes” or “no” during the intraoperative period.

### 2.8. Statistical Analysis

Statistical analysis was carried out with SPSS version 26. (IBM SPSS Statistic; IBM Corp., New York, NY, USA). Results are presented as mean ± standard deviation. The normality of data distribution was evaluated with the Shapiro–Wilk test. Intragroup repeatedly measured variables were analyzed with a one-way ANOVA followed by a post hoc Tukey test for multiple comparisons. For intergroup comparison of measured variables, an independent *t*-test was applied. Rescue analgesia and hypotension were evaluated using the Fisher exact test. Significance was set at *p* < 0.05.

## 3. Results

A total of 38 female domestic short hair cats undergoing OVE were included in the study with 18 cats assigned to CG and 20 cats assigned to ScEG. Cats in groups CG and ScEG weighed 2.9 ± 0.62 and 3.5 ± 0.7 kg, respectively, and their age was unknown due to their provenance. Surgery duration was 23.6 ± 4 and 20.2 ± 3.4 min in CG and ScEG (*p* = 0.012), respectively. During anesthesia, cats were maintained breathing spontaneously; there were no events to involve the institution of mechanical ventilation.

In the CG group, the PTAi value decreased following the cutaneous incision at T1. However, significant differences were observed at T2, T3, and T4 with respect to the baseline values. The HR increased and was significantly higher than the baseline at T2 and T3. During the procedure, there were no significant changes in the SAP, MAP, and *f*_R_ values in the CG (Table 1).

The PTAi value in ScEG showed significant differences between the baseline and T1, T2, T3, and T4. On the other hand, the HR did not show significant differences during the surgical period. Throughout the procedure, SAP, MAP, and *f*_R_ remained unchanged in ScEG, as shown in Table 2.

### Comparison between Groups

Comparison between groups (CG and ScEG) did not show significant differences in PTAi (*p* > 0.05) (Figure 1).

HR was significantly higher in CG compared to ScEG in baseline (*p* = 0.002), T1 (*p* < 0.001), T2 (*p* < 0.001), T3 (*p* = 0.001), and T4 (*p* < 0.002) (Figure 2).

SAP was significantly higher in CG compared to ScEG in T1 (*p* = 0.031) and T2 (*p* = 0.007) and MAP in T1 (*p* < 0.038) (Figure 2).

The *f*_R_ was significantly higher in CG compared to ScEG in T2 (*p* = 0.039) (Figure 2), without difference between groups in other points.

Changes in hemodynamic parameters greater than 20% that resulted in rescue analgesia occurred in 10 cats in CG and 3 cats in ScEG (*p* = 0.009).

Hypotension during anesthesia, defined as SAP < 90 mmHg or MAP < 60 mmHg was detected in 4 cats in CG and 11 cats in ScEG (*p* = 0.039). In all cases, hypotension was controlled with a reduction in isoflurane and crystalloid bolus.

## 4. Discussion

This randomized study suggests that in cats undergoing OVE, premedicated with a combination of dexmedetomidine and methadone, the administration of 0.3 mL kg^−1^ bupivacaine via ScE injection reduces intraoperative opioid consumption. However, the epidural administration of bupivacaine proved ineffective in maintaining PTAi levels between 50 and 70.

The notable disparities observed in the monitored hemodynamic parameters, including HR, SAP, and *f*_R_, along with the subsequent variations in the need for rescue analgesia, signify a significant opioid-sparing effect associated with ScE administration of bupivacaine in response to surgical stimuli. As studies in cats have suggested, the increased efficacy of ScE injection is attributed to the ability of the administered drugs within the epidural space to inhibit the activation and transmission of afferent neuronal impulses to the central nervous system in response to nociceptive stimuli [1,3].

The ScEG and the CG exhibited notable alterations in HR, specifically during the traction and ligation phases of the ovarian pedicles (T2, T3)—a recognized point of heightened surgical stimuli during ovariectomy [21]. Nevertheless, neither group showed any discernible variations in SAP, MAP, or *f*_R_ during the intraoperative period. However, non-invasive blood pressure measurements were only recorded every three minutes and may have missed some of the immediate changes in blood pressure, constraining the outcomes of the systolic and mean arterial blood pressure.

Three cats in the ScEG required rescue analgesia during surgery due to a hemodynamic variables shift exceeding 20%. Although no studies directly examine the innervation of feline ovaries, research conducted on dogs indicates that the sensory fibers providing innervation of ovaries may stem from the medullary segments between T10 and L4 [1]. According to the study by [22], injecting 0.1 to 0.3 mL kg^−1^ of new methylene blue into the lumbosacral epidural space in cats can reach T7 to T11 vertebrae.

Several factors may explain why ScE did not abolish the PTAi decrease. These include the dosage of bupivacaine administered, potential variations in epidural distribution compared to new methylene blue, individual differences in distribution, or inadequate cranial spread of the anesthetic solution to block medullary segments [1]. Additionally, the more caudal administration of bupivacaine, in contrast to the new methylene blue study, may contribute to these observations. The concentration of the drug affects the density of the nerve block. The higher the concentration, the more profound the motor and sensory nerve block [23]. This could be one reason for the sympathetic activation.

Regarding activation of sympathetic nervous system monitoring by the PTA monitor, significant differences were observed in PTAi during surgical stimuli in both groups. When comparing the PTAi values between groups, no significant differences were found. In cats, it has been suggested that a PTA values threshold value ≤ 51 is associated with the activation of the sympathetic nervous system [18]. In dogs, PTA was similar to cats [16], and targeted ANI values in adequately anesthetized humans, in a range between 50 and 70 [11]. Analyzing our PTAi results carefully is important since the threshold value is extrapolated from other species. Additionally, the effect of the drugs used on this threshold value is not fully understood. In our current study, baseline PTAi values above 50 were observed in all the cats before the onset of surgical stimuli. The mean PTAi values consistently remained below 50 throughout the study except for T1 in CG. This pattern suggests a potential activation of the sympathetic nervous system. Moreover, consistent correspondence between the reduction in PTAi below 50 and elevated values in hemodynamic variables or a significant increase (i.e., >20%) prompting the administration of rescue analgesia was not observed.

Previous studies in dogs utilizing PTA signaling demonstrated significantly decreased values associated with nociception but did not necessarily coincide with noteworthy changes in hemodynamic variables, such as heart rate and blood pressure [14]. This inconsistency makes PTA a valuable tool for detecting sympathetic nervous system activation but introduces controversy regarding its ability to predict hemodynamic changes. Similar observations have been documented in children, where a swift decline in ANI values followed surgical stimuli when locoregional anesthesia proved ineffective [24].

Based on our findings, using bupivacaine 0.25% administered at 0.3 mL kg^−1^ via ScE did not prevent the decrease in PTAi values in cats undergoing OVE. This supports the idea that nociception monitors could be useful in detecting sympathetic nervous system activation even in situations where locoregional anesthesia is applied [24]. It will be interesting in future studies to administer a higher volume via ScE to determine if sympathetic activation would be different.

Recent studies in rats have shown that ovaries receive communication through the vagus nerve [25]. The vagus nerve plays a crucial role in transmitting nociceptive information from visceral and somatic pain to the central nervous system [26]. Vagus nerves from cats and dogs [27] and humans [28] contain a sympathetic component, and vagus nerve stimulation may activate not only parasympathetic nerve fibers but also sympathetic nerve fibers [28,29]. Vagal afferent impulses may affect pain perception by indirectly activating the paraventricular nucleus, which then leads to changes in adrenaline release from the adrenal medulla. This phenomenon has been observed in rabbits [30]. These different sympathetic pathways may activate the sympathetic system detected by the PTA monitor, but they may not be enough to produce significant hemodynamic changes, as was seen in the GC at surgical time T4 in which the decrease in PTAi was not accompanied by significant changes in HR.

Related to PTAi values, an unexpected result was found. The mean PTAi values in the ScEG were generally lower than those in the CG. In a human study on patients under spinal anesthesia, the ANI value significantly decreased on average in 9 min post-spinal anesthesia when SAP dropped by 20% or fell below 100 mmHg [31]. Interestingly, the traction applied to the ovarian ligament creates tension in the parietal peritoneum—an organ seamlessly interconnected [32]. This traction, localized in the ovarian ligament region, could activate nociceptors positioned cranially beyond the uppermost reach of the nerve block, thus contributing to the observed PTAi response. However, the blockade of nerves supplying the adrenal gland (i.e., T13-L2) [33,34] attenuated the hemodynamic response. Nevertheless, further research is crucial to elucidate the complexities underlying the apparent discrepancy between hemodynamic fluctuations and PTAi assessments in cats receiving epidural analgesia.

Future studies should be carried out to reinforce our results and show the applicability of nociception monitors in feline anesthesia. There is a need to understand whether the activation of the sympathetic system, as detected by nociception monitors, specifically the PTA monitor, solely identifies nociception or if it is influenced by other mechanisms and pathways activating the sympathetic nervous system unrelated to nociception. This raises questions about the adequacy of currently available monitors in accurately assessing nociception or effectively regulating intraoperative opioid administration, as indicated by study of human anesthesia [35].

This study has some important limitations. The behavior associated with feral or stray cats in the present study precluded the collection of postoperative data. Postoperative pain evaluation would have been valuable to assess and compare the efficacy and duration of the regional anesthetic technique and determine potential advantages. Another important limitation was that the cutoff value used was extrapolated from studies in other species. Being a study and clinical context that seeks to demonstrate the effect of ScE on nociception and activation of the sympathetic nervous system, it should be considered that factors such as lack of independence between surgical stimuli and, in particular, the use of rescue analgesia with fentanyl may have influenced and introduced bias in our results since rescue analgesia was significantly higher in GC. Administering dexmedetomidine and methadone as pre-medication impacted the hemodynamic and sympathetic nervous response. The absence of invasive blood pressure monitoring precluded continuous and real-time assessment of the hemodynamic impact of the surgical procedure, which may have affected the results and statistical significance of SAP and MAP.

## 5. Conclusions

Our findings lead us to conclude that the sacrococcygeal epidural of 0.25% bupivacaine when compared to systemic analgesia alone could prevent an increase in hemodynamic variables (e.g., HR, SAP, and MAP). However, it is related to higher hypotension and does not prevent sympathetic nervous system activation. Also, the PTA monitor was not effective in differentiating nociceptive input between the two groups.

Future studies are necessary to evaluate the success rate and risks of sacrococcygeal epidural in feline ovariectomy and to understand the true role of activation of the sympathetic nervous system during anesthesia.

## Figures and Tables

**Figure 1 animals-14-01732-f001:**
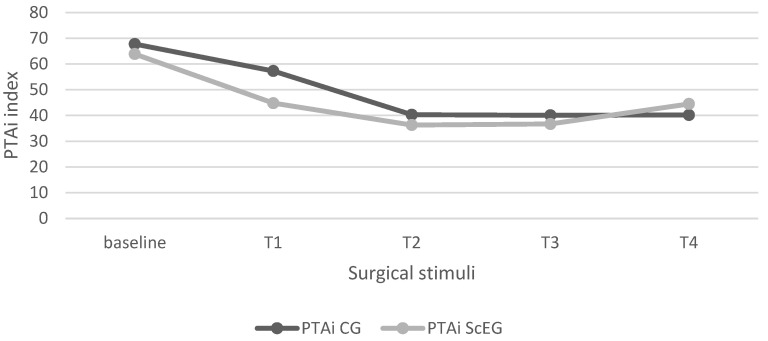
Mean PTAi values in CG and ScEG during surgical times (baseline, T1, T2, T3, and T4). CG = control group; PTAi = instantaneous parasympathetic tone activity; ScEG = sacrococcygeal epidural group.

**Figure 2 animals-14-01732-f002:**
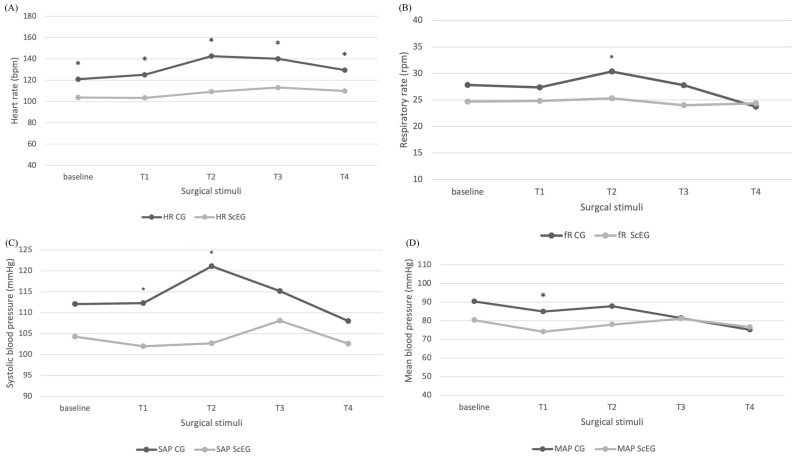
Monitoring values during five surgical times (baseline, T1, T2, T3, and T4). (**A**) Mean HR values in CG and EG; (**B**) mean *f*_R_ values; (**C**) mean values for systolic blood pressure; and (**D**) median blood pressure in CG and EG during surgical times. CG = control group; *f*_R_ = respiratory rate; HR = heart rate; MAP = median blood pressure; SAP = systolic blood pressure; ScEG = sacrococcygeal epidural group. * Significantly different between groups (*p* < 0.05).

**Table 1 animals-14-01732-t001:** Control group. Mean ± standard deviation for PTAi, HR, SAP, and MAP, *f*_R_ during surgery (baseline, T1, T2, T3, and T4). HR = heart rate; *f*_R_ = respiratory rate; PTAi = instantaneous parasympathetic tone activity; MAP = median arterial pressure; SAP = systolic arterial pressure.

	Baseline	T1	T2	T3	T4	*p*-Value
PTAI	68 ± 11 ^a^	57 ± 18 ^a^	40 ± 13 ^b^	40 ± 10 ^b^	40 ± 11 ^b^	<0.001
HR	121 ± 16.6 ^a^	125.1 ± 15.9 ^a^	143.6 ± 17.7 ^b^	140 ± 20.3 ^b^	129.4 ± 19.4 ^a^	0.002
SAP	113.1 ± 15.8 ^a^	112.3 ± 13.3 ^a^	121.1 ± 19.2 ^a^	115.2 ± 19.8 ^a^	108 ± 18.7 ^a^	0.321
MAP	90.4 ± 20 ^a^	84.9 ± 17.2 ^a^	84.8 ± 16.1 ^a^	81.5 ± 18.7 ^a^	75.1 ± 16 ^a^	0.173
*f* _R_	27.8 ± 8.9 ^a^	27.4 ± 8.2 ^a^	30.4 ± 8 ^a^	27.8 ± 8.2 ^a^	23.7 ± 6.9 ^a^	0.242

Different superscripts (a) and (b) indicate significant differences among surgical times (*p* < 0.05). The same superscripts indicate no significant differences observed between surgical times.

**Table 2 animals-14-01732-t002:** Sacrococcygeal epidural group. Mean ± standard deviation for PTAi, HR, SAP, and MAP, *f*_R_ during surgery (baseline, T1, T2, T3 and T4). HR = heart rate; *f*_R_ = respiratory rate; PTAi = instantaneous parasympathetic tone activity; MAP = median arterial pressure; SAP = systolic arterial pressure.

	Baseline	T1	T2	T3	T4	*p*-Value
PTAI	64 ± 10 ^a^	45 ± 19 ^b^	36 ± 13 ^b^	37 ± 15 ^b^	45 ± 17 ^b^	<0.001
HR	103.8 ± 15.4 ^a^	103.4 ± 11.9 ^a^	109.2 ± 13.2 ^a^	113.1 ± 14.5 ^a^	109.9 ± 14.4 ^a^	0.159
SAP	104.3 ± 14.9 ^a^	102.3 ± 13.8 ^a^	102.7 ± 18.9 ^a^	108.1 ± 12.4 ^a^	102.6 ± 13.1 ^a^	0.716
MAP	80.3 ± 18 ^a^	74.1 ± 12.6 ^a^	78 ± 16.8 ^a^	81.8 ± 13.9 ^a^	76.6 ± 12.9 ^a^	0.540
*f* _R_	24.7 ± 3.9 ^a^	24.8 ± 4.9 ^a^	25.3 ± 6 ^a^	24 ± 6.5 ^a^	24.4 ± 7.4 ^a^	0.970

Different superscripts (a) and (b) indicate significant differences among surgical times (*p* < 0.05). The same superscripts indicate no significant differences observed between surgical times.

## Data Availability

Data are contained within the article.

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
