# Peer review of "Can a Sacrococcygeal Epidural of 0.25% Bupivacaine Prevent the Activation of the Sympathetic Nervous System during Feline Ovariectomy?"

_animals, 2024, doi:10.3390/ani14121732_

Round 1

Reviewer 1 Report

Comments and Suggestions for Authors Well, to begin with, I would like to thank the authors for the effort made in preparing this manuscript. They have used an important series of quite objective parameters for the assessment of nociception, however, I have several considerations against it and I do not consider that the study method is appropriate, nor does it seem to me to be a clinically relevant study.     The first consideration is about the number of patients selected in the ScEG, as the previos power calculation estimated 18 cats per group. From my point of view, a Surgery Technique section seems to be essential in Material and Methods, as you do not explain if you had used a vessel sealing device or not. Also I think the extension of the abdominal incision is important. On the other hand, for this surgical time, I believe that a mean of 20 min to an experimented surgeon performe an ovariectomy in a cat is an little excessive time (so my previous question about the use or not of a vessel sealing). Another critical point is the anaesthetic protocol; in my opinión it could be more interesting the use of a control group in which a free opioids protocol was included, and in the ScEG avoid the use of them too. That could provides you more information about the analgesic effect of bupivacaine sacrococcygeal injection. Also you could avoid the hypontension in both of the groups. I don´t realy understand how you need an analgesic rescue with Fentanyle in that kind of interventions, in my experience it is unbelievable.   Despite the discussion of the results is well argumented, my apologies but I must decline the manuscript for its publication, so that I can´t find a clinical relevance as you mentioned that it is in a clinical context.

Author Response

Reviewer 1

Comments

Well, to begin with, I would like to thank the authors for the effort made in preparing this manuscript. They have used an important series of quite objective parameters for the assessment of nociception, however, I have several considerations against it and I do not consider that the study method is appropriate, nor does it seem to me to be a clinically relevant study.  

R:The first consideration is about the number of patients selected in the ScEG, as the previos power calculation estimated 18 cats per group. 

AR: Thank you for your comment. The sample size calculation indicated a minimum of 18 animals per group. Additional data were collected if necessary to account for excluded animals. The inclusion of 2 extra animals in one group has a negligible impact on type I and type II errors.

R: From my point of view, a Surgery Technique section seems to be essential in Material and Methods, as you do not explain if you had used a vessel sealing device or not.

AR: Thank you for your comment. The main objective of the present study is to evaluate the effect of sacrococcygeal epidural anesthesia on the sympathetic nervous system. The surgical technique, as described in our article, was performed by two experienced surgeons through median laparotomy. Since the purpose of this study is not to demonstrate the advantages of one surgical technique over another, it is the authors' opinion that a chapter dedicated to the description of the surgical technique does not bring any advantage to the understanding of the present study.

R: Also I think the extension of the abdominal incision is important. 

AR: Thank you for your comment. Given that both surgeons performed surgeries in both groups, there was no great variability that could affect the results of our study, particularly in terms of the extension of the abdominal incision.

R: On the other hand, for this surgical time, I believe that a mean of 20 min to an experimented surgeon performe an ovariectomy in a cat is an little excessive time (so my previous question about the use or not of a vessel sealing). 

AR: Thank you for your comment. As we described in our article, before the surgeon proceeded to the next surgical stimulus, it was expected that the PTAi would return to values between 50-70. This waiting time to obtain PTAi values ​​above 50-70, before moving on to the next surgical point, ended up increasing the duration of surgery time.

R: Another critical point is the anaesthetic protocol; in my opinión it could be more interesting the use of a control group in which a free opioids protocol was included, and in the ScEG avoid the use of them too. That could provides you more information about the analgesic effect of bupivacaine sacrococcygeal injection. Also you could avoid the hypontension in both of the groups.

AR: Thank you for your comment. As described in the objectives of our work, the main purpose of our study was to evaluate the effect of epidural sacrococcygeal anesthesia on the sympathetic system. The study was not designed to evaluate the effect of opioid-free anesthesia. On the other hand, a control group without opioids and locoregional anesthesia would jeopardize animal welfare and would be ethically reprehensible.

R: I don´t realy understand how you need an analgesic rescue with Fentanyle in that kind of interventions, in my experience it is unbelievable.  

AR: Thank you for your comment. To respect animal well-being and regardless of the type of surgery, whenever signs of intraoperative nociception occur (HR> 20%, SAP or MAP >20), rescue analgesics (fentanyl or ketamine) should be applied. In a study involving cats undergoing ovariohysterectomies with only opioids in the analgesic protocol, the intraoperative percentage of rescue analgesia was around 76,5% (Goich M, Bascuñán A, Faúndez P, Siel D. Comparison of analgesic efficacy of tramadol, morphine and methadone in cats undergoing ovariohysterectomy. Journal of Feline Medicine and Surgery. 2024;26(3). doi:10.1177/1098612X231224662). This underscores the need for the addition of regional anesthesia in these procedures.

R: Despite the discussion of the results is well argumented, my apologies but I must decline the manuscript for its publication, so that I can´t find a clinical relevance as you mentioned that it is in a clinical context.  

AR:Thank you very much for your opinion. Currently, intraoperative nociception/analgesia control is based on hemodynamic parameters (HR, SAP, MAP, RR), however these parameters have several limitations. Currently, in both human and veterinary medicine, several studies seek to validate/study the application of nociception monitors for better dosing of intraoperative analgesics. This study seeks to evaluate the feasibility of applying this type of monitors in feline species, being only the second study carried out with this type of monitors in felines.

Reviewer 2 Report

Comments and Suggestions for Authors

Dear authors, the article “Can Sacrococcygeal Epidural of 0,25% Bupivacaine Prevent the Activation of the Sympathetic Nervous System During Feline Ovariectomy” treats a new interesting topic. Overall the paper is well written, without spelling errors and the statistical analysis is also well planned. However, there are important deficiencies that I would like to clarify. Specific comments are listed below.

Author Response

Reviewer 2

Dear authors, the article “Can Sacrococcygeal Epidural of 0,25% Bupivacaine Prevent the Activation of the Sympathetic Nervous System During Feline Ovariectomy” treats a new interesting topic. Overall the paper is well written, without spelling errors and the statistical analysis is also well planned. However, there are important deficiencies that I would like to clarify. Specific comments are listed below.

ABSTRACT:

R: Please, you should specify the dose of bupivacine 0,25% used (ml/kg). You should also introduced more detail about the PTA (Parasympatetic Tone Activity monitor) used in this study to assess activation of the sympathetic nervous system, having regard to its relevance in the results obtained. AR: Thank you for your comment. We changed the abstract as suggested.

  1. INTRODUCTION:

R: Please, implement the description of the bibliographical results obtained previously from different studies about the effectiveness sacrococcygeal epidural in the ovariectomy of queens, focus your attention about the dosages (mg/kg or ml/kg) used to obtain an adequate cranial spread of the local anaesthetic used, in order to garantee an optimal analgesia for this type of surgery.

AR: Thank you for your comment. We added the information suggested.

R: Please try to better explain and confirm with the current bibliography the reason for which you choose the sacroccygeal epidural, instead of an epidural anesthesia at the lumbosacral level (L7-S1), which would certainly have brought a complete analgesia level to the procedure.

AR: Thank you for your comment. We changed the sentence as suggested.

The choice of ScE epidural was aimed at reducing the risk of puncturing the subdural sac and/or spinal cone, taking into account the anatomy of the feline species. Cadaveric studies in dogs “The cranial spread of contrast was similar, independent of whether the epidural injection was performed in the LS or ScE intervertebral space. Current volume guidelines used for the LS approach may produce similar distribution patterns when the ScE approach is used.” Vesovski S, Makara M, Martinez-Taboada F. Computer tomographic comparison of cranial spread of contrast in lumbosacral and sacrococcygeal epidural injections in dog cadavers. Vet Anaesth Analg. 2019 Jul;46(4):510-515. doi: 10.1016/j.vaa.2019.02.007. Epub 2019 Mar 9. PMID: 31155379. Given the absence of cadaveric studies conducted in cats demonstrating their real distribution when epidural is performed at the level of the ScE, we end up assuming a similar distribution. Additionally, using standard parameters (HR, FR, SAP, and MAP), there are studies in cats undergoing ovariectomies where ScE appeared to be an adjuvant analgesic (Dourado et al., 2023; Dos-Santos, 2024). The study that compared LS and ScE in cats did observed perioperative differences (Zahra JOL, Segatto CZ, Zanelli GR, Bruno TDS, Nicácio GM, Giuffrida R, Cassu RN. A comparison of intra and postoperative analgesic effects of sacrococcygeal and lumbosacral epidural levobupivacaine in cats undergoing ovariohysterectomy. J Vet Med Sci. 2023 Nov 2;85(11):1172-1179. doi: 10.1292/jvms.23-0114. Epub 2023 Oct 4. PMID: 37793832; PMCID: PMC10686773.)

R: From the bibliography that you have inserted at this point, there are no studies on the employment of the PTA in cats. Please describe it (if literature is present).

AR: Thank you for your comment. To the authors' knowledge, there is only one article on PTA monitoring in cats, which is referred to in the introduction. https://doi.org/10.3390/vetsci11030121

  1. MATERIALS AND METHODS:

R:

  • ANIMALS The dose (ml/kg) of bupivacaine 0,25% used shall be justified with appropriate bibliography. Why was not a larger volume used to increase the cranial spread of the local anesthetic? Having regard to the reduced concentration of the drug, even with twice the dosage the toxic dose in cats would not have been reached.

AR: Thank you for your comment. The concentration of 0.25% was used with the aim of prolonging analgesia without affecting motor function in the postoperative period. As for the volume, the chosen volume took into account both cadaveric (Lee et al. 2024) and in vivo studies (Dos-Santos et al. 2024), which, according to these sources, seemed sufficient to reach the necessary spinal segments (T7 to T11) promote analgesia in cat ovariectomies. Moreover, using a larger volume may lead to cranial migration, resulting in total or partial paralysis of the diaphragm and consequent increased respiratory depression.

  • INTRAOPERATIVE ANALGESIA AND NOCICEPTION          ASSESSMENTS                                                 AND RESCUE ANALGESIA

R: Please, insert the acronym of end-tidal Isoflurane (Etlso, %)

AR: Thank you for your comment. We have inserted the acronym in the sentence.

R: Since the objective of your study is a hemodynamic evaluation, why was a non-invasive method chosen for measuring pressure, given the unreliability of this method in cats? An invasive pressure measurement would have been more truthful and obtained in real time.

AR: Thank you for your comment. This fact constitutes a limitation of our study, which is mentioned in the discussion. Given the fact that the study was performed with feral cats included in a sterilization program. The surgical procedures were carried out in 2 days, which would make monitoring invasive blood pressure a more time-consuming and difficult process. Also, the cats were relatively small animals, which made it difficult to place and maintain an arterial catheter during all surgery.

R: How did you assess the depth of anaesthesiologic plan? It is necessary to specify this aspect since a superficiality of the anaesthetic plan can lead to a sympathetic stimulation, modifying the hemodynamic parameters detected in this study.

AR: Thank you for your comment. We have introduced a new sentence with the recommended information.

  • DATA COLLECTION

R: Delete "after" which is written twice ("after 15 minutes following epidural injection")

AR: Thank you for your comment. We have corrected the sentence.

RESULTS

R: Etlso are not taken into account in the results. Patients are maintained with isoflurane in pure oxygen in spontaneous breathing, so the work of breathing can be affected as well as the intake of isoflurane by the patient and, consequently, the depth of the anaesthesiologic plan. Please specifies the Etlso at different study time in both groups.

AR: Thank you for your comment. EtIso was maintained at 1 ± 0.1% throughout the surgery at all surgical times. There was only a temporary reduction of 20% in isoflurane in the cases where hypotension occurred, however, the anesthetic depth was always controlled by hands-on assessment (palpebral reflex, jaw tone, eye rotation).

R: In CG group, how you justified that at T4 there is a significant decrease in heart rate (HR) compared to T2-T3, which is not equivalent to a return of the normal ranges of PTAi? Was there a persistence of the sympathetic stimulation? Clarify it in the discussion please.

AR: Thank you for your comment. The only possible explanation is the activation of the sympathetic nervous system, and this stimulation of the sympathetic nervous system was not associated with nociceptive stimulation. We introduced a new sentence in the discussion to clarify this result.

R: In CG group, how is possible that at time T2 and T3, there is a decrease in PTAi and a statistically significant increase in HR and not an increase in MAP (mean arterial pressure), if the reason for these variations is a sympathetic stimulation? How do you justify this aspect? Clarify it in the discussion please.

AR: Thank you for your comment. The possible explanation for these results may be related to the use of non-invasive blood pressure, as mentioned in the limitations of our study, in the discussion section. We have improved the last sentence of the discussion to clarify this limitation.

  • COMPARISON BETWEEN GROUPS

R: In the CG group already at baseline, HR and MAP were significantly higher than the ScEG group. Is it possible that the depth of the anaesthetic plan affected these assessments? This assumption is also supported by the fact that the baseline time, ScEG group had a reduced RR (respiratory rate) compared to the C group, probably due to a deeper anaesthesiologic plan. Justify it please.

AR: Thank you for your comment. The same systemic anesthetic protocol was applied in both groups and animals were maintained in a surgical plan controlled by hands on monitoring. Given that in both groups baseline HR, MAP, SAP, and RR data were collected 15 minutes after the animals arrived in the operating room. The most likely explanation for the lower values ​​of HR, RR, SAP and MAP in ScEG is the sympathetic blockade promoted by ScE and not an anesthetic plan.

R: Please, specify in which study times rescue analgesia (fentanyl 2 µg/kg) was administered in both groups. Could the administration of fentanyl have influenced, from hemodynamic point of view, your evaluations? Why patients who received fentanyl were not ruled out from the study given the hemodynamic impact of this drug? Have you respected the half-life of fentanyl before recording subsequent assessments? Clarify it please.

AR: Thank you for your comment. Taking into account that this is a clinical study, we aimed to replicate a real clinical scenario as closely as possible. Therefore, by excluding animals receiving rescue interventions, we would lose valuable information on the hemodynamic parameters and PTA behavior in those situations. As this reflects a clinical scenario, the results obtained in this study reflect the hemodynamic and PTA behavior in this context.

Fentanyl was administered at times T2 and T3 in both groups. The half-life of fentanyl before subsequent assessments was not respected, new assessments were only initiated when PTA was between 50-70. Of course, fentanyl, due to its hemodynamic effects, may have affected our results, however, HR, RR, SAP and MAP values ​​were always higher in CG which was the group that received more rescue analgesia. The exclusion of animals that received fentanyl would imply results that would not reflect a real anesthetic context. The application of fentanyl as rescue analgesia was a limiting factor in our study and is mentioned in the discussion.

R: Can the reduction of isoflurane in 4 cats in CG group and 11 in the ScEG group, have influenced the evaluation and the sympathetic stimulation? Please, express your point of view.

AR: Thank you for your comment. That remains a hypothesis that we cannot entirely dismiss. However, in all cases, even after reducing ISO, the maintenance of the anesthetic plane was ensured, potentially subjectively eliminating the possibility of intraoperative awakening and consequent sympathetic activation.

  1. DISCUSSION

R: How do you justify that there are no significant differences in PTAi but there are hemodynamic differences (HR, SAP, MAP) between groups? Can PTAi not be a valid indicator for the detection of sympathetic stimulation in cats?

AR: Thank you for your comment. Within each group there was always activation of the sympathetic nervous system, however, there were no statistically significant differences between groups. Based on the results of our study and bibliography (Lima et al. 2024), the PTA monitor is very sensitive in detecting activation of the sympathetic nervous system in cats. However, taking into account other possible sympathetic pathways mentioned in the discussion, the PTA monitor in our study may have detected sympathetic activation that was not due to nociception or was associated with weak nociceptive stimuli, resulting in a decrease in PTAi values ​​that did not result in increases in HR, RR, SAP and MAP. In other words, possibly not all sympathetic stimuli detected by the PTA are nociceptive or have sufficient intensity to alter hemodynamic parameters. The same question has been raised in studies with ANI in human medicine, however, there are no studies related to this subject in cats.

R: Having regard to the uncertainty of the effectiveness of analgesia given by the sacrococcigeal epidural in the ovariectomy of the cats why you did not prefer a lumbosacral epidural for the evaluation of PTAi or a "splash block" with lidocaine 2%?

AR: Thank you for your comment. The choice of the sacrococcygeal epidural technique during the design of our study was related to the reduction of risks compared to the lumbosacral epidural in cats (Dourado et al. 2023, Otero et al. 2015). There are studies showing the analgesic efficacy of epidural sacrococcygeal anesthesia in cats undergoing ovariectomy, considering only the “gold standard” (FR, HR, MAP and SAP)(Dourado et al. 2023, Dos-Santos et al. 2024 and Zahra et al. 2023). Compared to the sacrococcygeal epidural, the splash block in abdominal surgery, particularly in OVE, is a more unpredictable technique, as with the latter it is impossible to guarantee that the entire visceral surface is blocked.

  1. CONCLUSIONS

R: Please change the sentence "Our finding led us to conclude the sacrococcygeal epidural of 0,25% bupivacaine when compared to systemic analgesia alone, provides more stability ... " to "Our finding led us to conclude the sacrococcygeal epidural of 0,25% bupivacaine when compared to systemic analgesia alone, could provide more stability ... "

AR: Thank you for your comment. We have changed the sentence.

Reviewer 3 Report

Comments and Suggestions for Authors

The title should have a question mark at the end of it.

The summaries state that the sacrococcygeal epidural resulted in ‘greater hemodynamic stability’. This does not seem to be a valid conclusion since 55% of this group suffered from hypotension?

‘Two groups of animals were evaluated in five peri-operative moments’ – please use ‘times’ rather than moments.

‘PTA values nearing 100 denote a predominant parasympathetic tone or opioid overdose, while values below 50 denote a predominant sympathetic tone associated with elevated stress or nociception’ It should be noted here that these values have not been validated in cats – your own study used these values but did not validate the cutoff points. I think it would be helpful to describe how these numbers are derived – what kind of transformation is used to get these values?

‘To achieve 80% power for detecting a medium effect’ – please define the criteria you used for a ‘medium effect’.

‘were premedicated via intramuscular (IM) dexmedetomidine and methadone’ – it should be noted somewhere that this premedication would tend to increase parasympathetic tone.

oxygen haemoglobin saturation – should be arterial oxygen…

You measured end-tidal iso and CO2 – please describe how the monitor was calibrated.

number 2 cuff – please define dimensions – a #2 cuff from one manufacturer may not be the same as from another.

‘Rescue analgesia consisted of an IV bolus of fentanyl’ – how did you handle those animals that received the fentanyl. This would tend to increase parasympathetic tone so shouldn’t these animals be eliminated from further analysis?

‘commenced only after after 15 minutes ‘ – eliminate one after.

Table 1 and 2. Please express these values to the accuracy of the measurement. All values and SDs should be whole integers. The figures repeat what is in the tables so please use one or the other.

‘The notable disparities noted in monitored hemodynamic parameters, including HR, SAP, and fR’ The latter is not a ‘hemodynamic parameter’. I think it is difficult to interpret the lower blood pressure as being an antinociceptive effect when we know that epidural local anesthetics will block sympathetic tone which may reduce blood pressure. The fact that fR was different at one time point is hardly a ‘notable’ difference.

‘As posited studies in cats,’ I am not sure I understand what this means – please rephrase.

‘However, it is crucial to acknowledge that the nature of non-invasive blood pressure measurements may constrain the outcomes of systolic and mean arterial blood pressure.’ Is this just saying that these measurements are unreliable or that you don’t know how the measurements change with acute changes in pressure or …? Please be specific.

‘In cats it has been suggested that a PTA value ≤51…’ This is your own work and you did not validate this value.

In the last paragraph you detail study limitations. The first major limitation is that the values you are using for the PTA are assumed. You have no evidence that the cutoff values are valid in cats. You also do not know the influence of the drugs you used on these values. As indicated above, both dexmedetomidine and methadone would tend to increase parasympathetic tone so would this shift the cutoff values? You also have a complicating factor with the epidural in that it clearly affects sympathetic tone so these animals might have a different autonomic balance just due to the epidural. 

Comments on the Quality of English Language

Some grammatical errors

Author Response

Reviewer 3

R: The title should have a question mark at the end of it.

AR: Thank you for your comment. We have changed the title.

R: The summaries state that the sacrococcygeal epidural resulted in ‘greater hemodynamic stability’. This does not seem to be a valid conclusion since 55% of this group suffered from hypotension?

AR: Thank you for your comment. We have changed the sentence. We wanted to state that it prevents an increase in parameters associated with analgesic rescue (HR, RR, SAP, MAP). The hypotension recorded was mild and responded favorably to the reduction of isoflurane and/or crystalloids.

R: ‘Two groups of animals were evaluated in five peri-operative moments’ – please use ‘times’ rather than moments.

AR: Thank you for your comment. We have changed the sentence.

R: ‘PTA values nearing 100 denote a predominant parasympathetic tone or opioid overdose, while values below 50 denote a predominant sympathetic tone associated with elevated stress or nociception’ It should be noted here that these values have not been validated in cats – your own study used these values but did not validate the cutoff points. I think it would be helpful to describe how these numbers are derived – what kind of transformation is used to get these values?

AR: Thank you for your comment. Currently, no study has validated the cutoff values ​​in the feline species, and these values ​​have been extrapolated from studies carried out in humans, dogs, and pigs that used these values. In a previous study carried out in cats (https://doi.org/10.3390/vetsci11030121), we verified the superiority of PTA in detecting activation of the sympathetic nervous system when compared to hemodynamic parameters (HR, RR, SAP and MAP).

R: ‘To achieve 80% power for detecting a medium effect’ – please define the criteria you used for a ‘medium effect’.

AR: Thank you for your comment. A medium effect is related to the detection of approximately a difference of 25 bpm HR or 25 mmHg in SAP and MAP between groups.

R: ‘were premedicated via intramuscular (IM) dexmedetomidine and methadone’ – it should be noted somewhere that this premedication would tend to increase parasympathetic tone.

oxygen haemoglobin saturation – should be arterial oxygen…

AR: Thank you for your comment. We understand your comment. However, in our view, any drug used in the anesthetic protocol would affect the activity of the autonomic nervous system.

R: You measured end-tidal iso and CO2 – please describe how the monitor was calibrated.

AR: Thank you for your comment. The calibration was performed by equipment technicians (Mindray) using a gas analyzer

R: number 2 cuff – please define dimensions – a #2 cuff from one manufacturer may not be the same as from another.

AR: Thank you for your comment. The dimensions are 3.1 to 5.7cm and were added in the main document.

R: ‘Rescue analgesia consisted of an IV bolus of fentanyl’ – how did you handle those animals that received the fentanyl. This would tend to increase parasympathetic tone so shouldn’t these animals be eliminated from further analysis?

AR: Thank you for your comment. This was a study in a clinical context. If, on the one hand, we seek to evaluate the effect of sacrococcygeal anesthesia on the sympathetic nervous system, also seek to evaluate the feasibility of using the monitor in both groups.

Fentanyl was administered at times T2 and T3 in both groups. The half-life of fentanyl before subsequent assessments was not respected, new assessments were only initiated when PTA was between 50-70. Of course, fentanyl, due to its hemodynamic effects, may have affected our results, however, HR, RR, SAP and MAP values ​​were always higher in CG which was the group that received more rescue analgesia. The exclusion of animals that received fentanyl would imply results that would not reflect a real anesthetic context. The application of fentanyl as rescue analgesia was a limiting factor in our study and is mentioned in the discussion.

R:‘commenced only after after 15 minutes ‘ – eliminate one after.

AR: Thank you for your comment. We have changed the sentence.

R: Table 1 and 2. Please express these values to the accuracy of the measurement. All values and SDs should be whole integers. The figures repeat what is in the tables so please use one or the other.

‘The notable disparities noted in monitored hemodynamic parameters, including HR, SAP, and fR’ The latter is not a ‘hemodynamic parameter’. I think it is difficult to interpret the lower blood pressure as being an antinociceptive effect when we know that epidural local anesthetics will block sympathetic tone which may reduce blood pressure. The fact that fR was different at one time point is hardly a ‘notable’ difference.

AR: Indeed, the data were analyzed twice. However, they were evaluated in two ways: intergroup and intragroup. This way, it's possible to analyze the behavior at different moments within each group, but also to compare the behavior between the two groups.

R:‘As posited studies in cats,’ I am not sure I understand what this means – please rephrase.

AR: Thank you for your comment. We rephrase as suggested.

R: ‘However, it is crucial to acknowledge that the nature of non-invasive blood pressure measurements may constrain the outcomes of systolic and mean arterial blood pressure.’ Is this just saying that these measurements are unreliable or that you don’t know how the measurements change with acute changes in pressure or …? Please be specific.

AR: Thank you for your comment. This fact constitutes a limitation of our study, which is mentioned in the discussion. Given the fact that the study was performed with feral cats included in a sterilization program and monitoring invasive blood pressure a more time-consuming and difficult process for animals with ASA I. Also, the cats were relatively small animals, which made it difficult to place and maintain an arterial catheter during all surgery. Taking into account that non-invasive pressure measurements occur every 3 minutes, instantaneous variations may not be observed.

R:‘In cats it has been suggested that a PTA value ≤51…’ This is your own work and you did not validate this value.

AR: Although not strictly a validation study, the only work available in cats used this value (Lima et al. 2024). However, the values used in other species, humans, dogs, and pigs, show no significant variation between species in the cutoff value considered.

R: In the last paragraph you detail study limitations. The first major limitation is that the values you are using for the PTA are assumed. You have no evidence that the cutoff values are valid in cats. You also do not know the influence of the drugs you used on these values. As indicated above, both dexmedetomidine and methadone would tend to increase parasympathetic tone so would this shift the cutoff values? You also have a complicating factor with the epidural in that it clearly affects sympathetic tone so these animals might have a different autonomic balance just due to the epidural. 

AR: Thank you for your comment. These limitation was added.  

R:Some grammatical error

AR: Thank you for your comment. The article has been reviewed by a native English speaker

Round 2

Reviewer 1 Report

Comments and Suggestions for Authors

Well, I thank the authors of the manuscript for the answers obtained to the questions raised. Reevaluating the changes made to the article, I consider that it could be published although with minor changes. For me it is essential, since they continue to treat it with a clinical objective, to introduce a brief explanation of the surgical method, even though the authors do not consider it. It seems to me that, although there are no major differences between the animals in the study, since the surgeons are always the same, the anesthetic protocol is, as always, influenced by the surgical technique carried out. If we also consider its clinical purpose, a description seems essential to me, so that any clinician can extrapolate results.

Author Response

Reviewer 1

R: Well, I thank the authors of the manuscript for the answers obtained to the questions raised. Reevaluating the changes made to the article, I consider that it could be published although with minor changes. For me it is essential, since they continue to treat it with a clinical objective, to introduce a brief explanation of the surgical method, even though the authors do not consider it. It seems to me that, although there are no major differences between the animals in the study, since the surgeons are always the same, the anesthetic protocol is, as always, influenced by the surgical technique carried out. If we also consider its clinical purpose, a description seems essential to me, so that any clinician can extrapolate results.

AR: Thank you for your comment. We introduce a section in the work, briefly describing the surgical technique (2.3. Surgical Technique)

Reviewer 2 Report

Comments and Suggestions for Authors

Thank you for the changes made to your work. I really enjoyed to your article.

Author Response

Reviewer 2

R: Thank you for the changes made to your work. I really enjoyed to your article.

AR: Thank you for the excellent review that helped to improve our article greatly.

Reviewer 3 Report

Comments and Suggestions for Authors

Abstract

Please use hemodynamic variables not parameters.

 ‘Compared to systemic analgesia alone, sacrococcygeal epidural with 0.25% bupivacaine could provide more hemodynamic stability but does not prevent sympathetic nervous system activation in cats undergoing ovariectomy.’ Saying ‘could provide’ is meaningless. This sentence is effectively a repeat of the previous sentence so please delete.

Introduction

During ovariohysterectomy (‘the’ and ‘procedure’ are not needed)

Peritoneal pain – if the animal is anesthetized there is no pain because the latter is a conscious perception. Use ‘nociception’ instead.

‘and allows to complement or replace systemic analgesia’ – ‘and can complement or replace’

‘Sacrococcygeal epidural (ScE) anaesthesia is a modality of locoregional anaesthesia that promotes preventive perioperative analgesia’ – ‘Sacrococcygeal epidural (ScE) anaesthesia is a locoregional technique that prevents perioperative nociception.’ Can you use the term analgesia in an anaesthetized patient since the term means ‘no pain’  and so the same comment applies as above (please alter throughout).

‘However, BP and HR are not objective indicators of nociception and may be subject to confounding factors and not al-ways accurately reflect nociception [11,12].’ Part of this sentence is plagiarized directly from ref 11. That paper does not give a reference for this statement and the second reference cites issues that would only occur with laparoscopy and the Trendelenberg position. I don’t disagree with your statement but please find better references to support it that might apply under the conditions of this study.

You state that the ANI monitor is being increasingly used but a recent meta-analysis appears to show no benefit - Pain Assessment Using the Analgesia Nociception Index (ANI) in Patients Undergoing General Anesthesia: A Systematic Review and Meta-Analysis. 

Kim MK, Choi GJ, Oh KS, Lee SP, Kang H. J Pers Med. 2023 Oct 4;13(10):1461. doi: 10.3390/jpm13101461. The evidence they cite in this paper is weak so it would appear that there is very little scientific basis for using this approach. The results from some of the individual studies in this analysis appear to be all over the map. In Susano (2021) there is no decrease below 50 with tetanic stimulation at 1.5 ng/mL remifentanil whereas in Gruenwald (2013) the ANI decreases from 71 to 30 with the tetanic stimulation at 2 ng/mL remifentanil! 

The description of this monitor reminds me of a cartoon that showed a bunch of calculations then an arrow pointing to the words ‘then a miracle occurs’ and then an arrow pointing to the outcome. The conversion of heart rate variability to numbers between 0-100 must have some scientific basis and it would help the reader if you could give some description of how this occurs.  My assumption is that 50 represents the balance point between the sympathetic and parasympathetic tone, but that balance point must shift if you had drugs into the mix.

‘In anaesthetized patients, PTA values that fall within the range of 50 to 70, indicate the absence of nociception [13,15,16].’ I don’t think this is the absence of nociception but the absence of a nociceptive effect on the autonomic nervous system. If you give an opioid that blocks nociceptive input at the spinal cord and enhances descending inhibition, the nociceptors in the periphery are still firing to indicate that a noxious stimulus is present (as you state in the discussion).

A priori power calculation – please provide the criteria used – what difference were you expecting and what data did you base this on?

‘serum biochemistry and haematology were performed’ – were the results of these tests available before the cat was anaesthetized? If not, were they somehow analysed in retrospect to allocate or eliminate animals to/from the study?

3.1 to 5.1 cm cuff – you need to specify the width of the cuff (the length is less relevant).

‘a lead II ECG signal was used after plac-ing three flattened crocodile clips moistened with gel on the skin at the level of the olec-ranon of the right thoracic limb and over the patellar ligaments of the pelvic limbs’. Typically a lead 2 configuration would be the right thoracic limb and the left pelvic limb with the third lead on the left thoracic limb.

‘In case of hypotension’ – you don’t mention the management of bradycardia. Many of the cats in the epidural group appear to be bradycardic (HR<100 bpm in cats).

‘(T4) were recorded.’ You have already said ‘obtained’ so ‘were recorded’ is redundant.

Statistical analysis – aren’t the values obtained from the ANI effectively non-parametric? Does a value of 20 represent twice the nociceptive signal as a value of 40? Despite this scale being from 0-100 the relationships between the numbers are not known so the data should not be analyzed as if the values were arithmetic. This also means that the PTAi values should be reported as median/range.

Table 1&2 – as indicate before values should be reported as whole integers not to one decimal place.

‘However, it is crucial to acknowledge that the nature of non-invasive blood pressure measurements may constrain the outcomes of systolic and mean arterial blood pressure.’ This is obtuse – please state something like – because the noninvasive monitor only records values every three minutes it may have missed some of the immediate changes in blood pressure.

‘The concentration of the drug varies with the distance to the injection point which could imply a lesser effect in ScEG at the cranial abdomen [22].’ You are quoting a study that used intrathecal administration of drugs where dilution with CSF is likely. Do you have any similar data for epidural administration?

‘When comparing PTAi values between groups no significant differences were found. In cats, it has been suggested that a PTA values threshold value ≤ 51 is associated with the activation of the sympathetic nervous system [18]. In dogs, PTA was similar to cats [16], and targeted ANI values in adequately anaesthetized humans, range between 50 and 70 [11].’ As mentioned before – this needs to be expanded to state that there is no basis for the value of 51 in cats and that there is no understanding of the influence of drugs on this threshold value.

‘In our current study, baseline PTAi values above 50 were observed in all the cats before the onset of surgical stimuli.’ Was this a self-fulfilling prophesy – you mentioned that you required a value between 50-70 before proceeding so if this is what you used to monitor the animal and you adjusted anesthetic depth accordingly then that is why these values were observed?

‘This supports the idea that nociception monitors could be useful in detecting sympathetic nervous system activation even in situations where locoregional anaesthesia is applied [23].’ It could also be true that the activation of the sympathetic nervous system and nociception are different. In your ref 12 the patients had no difference in stress response when ANI was used.

‘In a human study on patients under spinal anaesthesia, the ANI value significantly decreased at 3 minutes post-spinal anaesthesia when SAP dropped by 20% or fell below 100 mmHg [30].’ Is this a reflection of the block or the change in body position? The changes were present by 3 minutes – is that enough time for bupivacaine to block the sympathetic system?

Administering dexmedetomidine and methadone as pre-medication might have impacted the hemodynamic and sympathetic nervous response. – ‘might have’? This does of dexmedetomidine would cause at least a 50% reduction in cardiac output and a 300% increase in systemic vascular resistance – there is no ‘might’ about it!

‘could provide more stability’ rather than using ‘stability’ I think you should state that it made the blood pressure and heart rate less likely to increase with nociceptive stimulation but that these animals required more intervention for a decreased blood pressure. I think that your conclusion should also state that PTAi was not effective in differentiating nociceptive input between the two treatments.

Comments on the Quality of English Language

There will need to be significant editing to make the language more concise

Author Response

Reviewer 3

Abstract

R: Please use hemodynamic variables not parameters.

AR: Thank you for your comment. We changed the words as suggested.

R: ‘Compared to systemic analgesia alone, sacrococcygeal epidural with 0.25% bupivacaine could provide more hemodynamic stability but does not prevent sympathetic nervous system activation in cats undergoing ovariectomy.’ Saying ‘could provide’ is meaningless. This sentence is effectively a repeat of the previous sentence so please delete.

AR: Thank you for your comment. We removed the sentence as suggested.

Introduction

R: During ovariohysterectomy (‘the’ and ‘procedure’ are not needed)

AR: Thank you for your comment. We removed the words as suggested.

R: Peritoneal pain – if the animal is anesthetized there is no pain because the latter is a conscious perception. Use ‘nociception’ instead.

AR: Thank you for your comment. We changed the word as suggested.

R: ‘and allows to complement or replace systemic analgesia’ – ‘and can complement or replace’

AR: Thank you for your comment. We changed the sentence as suggested.

R: ‘Sacrococcygeal epidural (ScE) anaesthesia is a modality of locoregional anaesthesia that promotes preventive perioperative analgesia’ – ‘Sacrococcygeal epidural (ScE) anaesthesia is a locoregional technique that prevents perioperative nociception.’ Can you use the term analgesia in an anaesthetized patient since the term means ‘no pain’  and so the same comment applies as above (please alter throughout).

AR: Thank you for your comment. We changed the sentence as suggested. We also change the terms throughout the article

R: ‘However, BP and HR are not objective indicators of nociception and may be subject to confounding factors and not al-ways accurately reflect nociception [11,12].’ Part of this sentence is plagiarized directly from ref 11. That paper does not give a reference for this statement and the second reference cites issues that would only occur with laparoscopy and the Trendelenberg position. I don’t disagree with your statement but please find better references to support it that might apply under the conditions of this study.

AR: Thank you for your comment. We have introduced a new bibliographic reference, as recommended.

R:You state that the ANI monitor is being increasingly used but a recent meta-analysis appears to show no benefit - Pain Assessment Using the Analgesia Nociception Index (ANI) in Patients Undergoing General Anesthesia: A Systematic Review and Meta-Analysis.

Kim MK, Choi GJ, Oh KS, Lee SP, Kang H. J Pers Med. 2023 Oct 4;13(10):1461. doi: 10.3390/jpm13101461. The evidence they cite in this paper is weak so it would appear that there is very little scientific basis for using this approach. The results from some of the individual studies in this analysis appear to be all over the map. In Susano (2021) there is no decrease below 50 with tetanic stimulation at 1.5 ng/mL remifentanil whereas in Gruenwald (2013) the ANI decreases from 71 to 30 with the tetanic stimulation at 2 ng/mL remifentanil!

AR: Thank you for your comment.  The use of this equipment (ANI) in human medicine has increased. There are many studies in human medicine and the results are not consensual regarding the use of this equipment. Given the information gap that exists in veterinary medicine regarding the use of PTA monitor, particularly in feline species, we carried out this study to verify its applicability and our conclusions are not too far removed from the conclusions obtained in human medicine, that is, this monitor does not only seem to detect nociception.

R: The description of this monitor reminds me of a cartoon that showed a bunch of calculations then an arrow pointing to the words ‘then a miracle occurs’ and then an arrow pointing to the outcome. The conversion of heart rate variability to numbers between 0-100 must have some scientific basis and it would help the reader if you could give some description of how this occurs.  My assumption is that 50 represents the balance point between the sympathetic and parasympathetic tone, but that balance point must shift if you had drugs into the mix.

AR: Thank you for your comment. We agree with your opinion. In fact, the middle value (50) reflects the balance between a higher percentage of sympathetic and parasympathetic activity. On the other hand, as we mentioned in the article, we have to base it on values already mentioned in other articles, which are actually in line with the value of 50 used in ours.

R: ‘In anaesthetized patients, PTA values that fall within the range of 50 to 70, indicate the absence of nociception [13,15,16].’ I don’t think this is the absence of nociception but the absence of a nociceptive effect on the autonomic nervous system. If you give an opioid that blocks nociceptive input at the spinal cord and enhances descending inhibition, the nociceptors in the periphery are still firing to indicate that a noxious stimulus is present (as you state in the discussion).

AR: Thank you for your comment. We agree with your statement, that according to the operating method of this equipment, nociceptive stimuli that do not alter heart rate variability will not be detected by the PTA.

The description made by us in the introduction is based on current literature, however, our results suggest that this equipment may not only detect nociceptive stimuli, but non-nociceptive activations of the sympathetic nervous system.

R: A priori power calculation – please provide the criteria used – what difference were you expecting and what data did you base this on?

AR: Thank you for your comment. In order to perform a priori power calculation, we used a criterion of a difference of 25 bpm in HR or 25 mmHg in SAP and MAP between groups. The effect size was estimated based on previous published data. For example, we considered the findings of Dourado et al. (2023), where they observed a notable difference in heart rate (HR) between the Control group and ScE group. Specifically, there was an approximate increase of 27 beats per minute (bpm) in the ScE group, representing a rise of just over 20% compared to the Control group.

R: ‘serum biochemistry and haematology were performed’ – were the results of these tests available before the cat was anaesthetized? If not, were they somehow analysed in retrospect to allocate or eliminate animals to/from the study?

AR: Thank you for your comment. Given the nature of these cats (feral cats), serum biochemistry and haematology were performed after premedication. Although the samples were carried out under sedation, the results were obtained before the surgical procedure began.

R: 3.1 to 5.1 cm cuff – you need to specify the width of the cuff (the length is less relevant).

AR: Thank you for your comment. The cuff width is 2.5 cm, we change the information in the article.

R: ‘a lead II ECG signal was used after placing three flattened crocodile clips moistened with gel on the skin at the level of the olecranon of the right thoracic limb and over the patellar ligaments of the pelvic limbs’. Typically a lead 2 configuration would be the right thoracic limb and the left pelvic limb with the third lead on the left thoracic limb.

AR: Thank you for your comment. We followed the manufacturer's recommendations regarding the application of the electrodes, we removed the incorrect information from the document

R: ‘In case of hypotension’ – you don’t mention the management of bradycardia. Many of the cats in the epidural group appear to be bradycardic (HR<100 bpm in cats).

AR: Thank you for your comment. We added atropine to the possible intra-operative management. It wasn't necessary to use atropine, because although the HR in the ScE group was very close to normal HR, the pressures were resolved with a reduction in isoflurane and/or fluid boluses.

R: ‘(T4) were recorded.’ You have already said ‘obtained’ so ‘were recorded’ is redundant.

AR: Thank you for your comment. We changed the sentence.

R: Statistical analysis – aren’t the values obtained from the ANI effectively non-parametric? Does a value of 20 represent twice the nociceptive signal as a value of 40? Despite this scale being from 0-100 the relationships between the numbers are not known so the data should not be analyzed as if the values were arithmetic. This also means that the PTAi values should be reported as median/range.

AR: Although PTAi values ​​can vary from 0-100, and it is impossible to say that an animal with 20 has twice the nociception of an animal with 40, the PTAi variable in statistical terms is quantitative or continuous, and studies carried out in human and veterinary medicine have approached PTAi values ​​in this way. Transforming a quantitative variable into a qualitative nominal or ordinal variable was possible, but with an ordinal variable, we would not have any support for creating ranges of values, which would make this approach completely random and unfounded.

R: Table 1&2 – as indicate before values should be reported as whole integers not to one decimal place.

AR: Thank you for your comment.  Although these are average values, we modified the PTAi values ​​to integer values ​​and not decimal values ​​as the equipment only displays integers.

R: ‘However, it is crucial to acknowledge that the nature of non-invasive blood pressure measurements may constrain the outcomes of systolic and mean arterial blood pressure.’ This is obtuse – please state something like – because the noninvasive monitor only records values every three minutes it may have missed some of the immediate changes in blood pressure.

AR: Thank you for your comment.  We changed the sentence as suggested.

R: ‘The concentration of the drug varies with the distance to the injection point which could imply a lesser effect in ScEG at the cranial abdomen [22].’ You are quoting a study that used intrathecal administration of drugs where dilution with CSF is likely. Do you have any similar data for epidural administration?

AR: Thank you for your comment. We removed this sentence, since the study rfers to  intrathecal administration of drugs, and we cannot transpose it to epidural administration

R: ‘When comparing PTAi values between groups no significant differences were found. In cats, it has been suggested that a PTA values threshold value ≤ 51 is associated with the activation of the sympathetic nervous system [18]. In dogs, PTA was similar to cats [16], and targeted ANI values in adequately anaesthetized humans, range between 50 and 70 [11].’ As mentioned before – this needs to be expanded to state that there is no basis for the value of 51 in cats and that there is no understanding of the influence of drugs on this threshold value.

AR: Thank you for your comment. We have introduced a new sentence to highlight this limitation regarding PTAi

R: ‘In our current study, baseline PTAi values above 50 were observed in all the cats before the onset of surgical stimuli.’ Was this a self-fulfilling prophesy – you mentioned that you required a value between 50-70 before proceeding so if this is what you used to monitor the animal and you adjusted anesthetic depth accordingly then that is why these values were observed?

AR: Thank you for your comment. The PTAi value was not used to manage the depth of anesthesia. In other words, the isoflurane value was not based on the PTAi value. What we indicated was that after a drop in PTAi below 50, following a nociceptive stimulus, the PTAi value was allowed to rise to values above 50, before starting the next "T".

R: ‘This supports the idea that nociception monitors could be useful in detecting sympathetic nervous system activation even in situations where locoregional anaesthesia is applied [23].’ It could also be true that the activation of the sympathetic nervous system and nociception are different. In your ref 12 the patients had no difference in stress response when ANI was used.

AR: Thank you for your comment. It means that in addition to the known nociceptive pathways, there may be other pathways that are poor understood or even that nociception monitors may be activated by other sympathetic pathways unrelated to nociception.

R: ‘In a human study on patients under spinal anaesthesia, the ANI value significantly decreased at 3 minutes post-spinal anaesthesia when SAP dropped by 20% or fell below 100 mmHg [30].’ Is this a reflection of the block or the change in body position? The changes were present by 3 minutes – is that enough time for bupivacaine to block the sympathetic system?

AR: Thank you for your comment. We changed the sentence, since the median value was 9 min (3-30 min). Also, in the cited study, bupivacaine was administered intrathecal.

R: Administering dexmedetomidine and methadone as pre-medication might have impacted the hemodynamic and sympathetic nervous response. – ‘might have’? This does of dexmedetomidine would cause at least a 50% reduction in cardiac output and a 300% increase in systemic vascular resistance – there is no ‘might’ about it!

AR: Thank you for your comment. We changed the sentence as suggested.

R: ‘could provide more stability’ rather than using ‘stability’ I think you should state that it made the blood pressure and heart rate less likely to increase with nociceptive stimulation but that these animals required more intervention for a decreased blood pressure. I think that your conclusion should also state that PTAi was not effective in differentiating nociceptive input between the two treatments.

AR: Thank you for your comment. We have modified the conclusions section to include the recommendations.